# MultiTimeSurv: Temporal Multimodal Networks for Dynamic Survival Analysis with Longitudinal Data

## Abstract

Survival analysis requires modeling complex temporal dependencies and multimodal data to predict outcomes accurately. Existing state-of-the-art methods, such as Dynamic-DeepHit, have advanced temporal survival modeling but remain constrained to tabular data and cannot leverage multimodal information, leaving critical gaps in handling irregular sampling, heterogeneous modalities, and cross-modal alignment. In this way, we introduce MultiTimeSurv, a novel deep learning framework that integrates longitudinal tabular data with image analysis for dynamic survival prediction. Our approach addresses three key challenges: (1) capturing temporal evolution through attention-based recurrent networks, (2) processing multimodal data via specialized feature encoders for tabular embeddings and a transformer-based image analysis module, and (3) handling missing data patterns common in real-world settings. MultiTimeSurv employs contextual embeddings for categorical and continuous variables, a temporal attention mechanism for longitudinal modeling, and a fully transformer-based architecture for extracting visual-textual features from images. We evaluate MultiTimeSurv on multiple datasets, including hospitalization data, longitudinal studies, and multimodal image-text datasets, outperforming the current state-of-the-art survival analysis methods. On SYMILE-MIMIC, it consistently surpasses classical and neural baselines across all horizons, exceeding a C-index of 0.70 at long-term predictions.

## 1 Introduction

Temporal multimodal data plays a central role in numerous applications, from medical prognosis to dynamic risk assessment. A particularly important setting is survival analysis, where the goal is to model time-to-event outcomes under the presence of censoring and competing risks (Cox, 1972; Kalbfleisch & Prentice, 2002; Wiegrebe et al., 2024). Unlike standard predictive tasks, survival modeling must contend with incomplete event information, irregular observation times, and heterogeneous input sources (Wang et al., 2019; Kvamme et al., 2019). These challenges make temporal multimodal survival analysis substantially more complex than conventional sequence learning (Lee et al., 2019; Wiegrebe et al., 2024).

Real-world datasets for survival analysis typically violate the assumptions of most deep learning architectures (Lee et al., 2019). Tabular time series mix categorical and continuous variables arriving at irregular intervals (Shukla & Marlin, 2021; Futoma et al., 2017); high-dimensional modalities such as medical imaging are sparsely sampled (Rajpurkar et al., 2022; Çallı et al., 2021); and temporal alignment across modalities is often noisy or missing entirely (Guarrasi et al., 2025). Existing neural architectures usually assume uniform sampling, homogeneous input types, and consistent temporal resolution (Zerveas et al., 2021; Lim et al., 2021), leading to poor robustness in practice (Harutyunyan et al., 2019).

Current approaches fall short in several key aspects. First, fusion strategies commonly rely on concatenation or early fusion, which ignore cross-modal interactions and temporal dependencies (Guarrasi et al., 2025; Tsai et al., 2019). Processing each modality independently before fusion discards valuable synchronization cues, resulting in suboptimal joint representations (Akbari et al., 2021).

Second, handling irregular temporal sampling remains a bottleneck: recurrent or transformer-based models typically assume fixed time steps (Vaswani et al., 2017), while naive imputation introduces artifacts (Shukla & Marlin, 2021). Third, survival analysis introduces its own algorithmic constraints. Classical methods such as the Cox model assume proportional hazards and static features (Cox, 1972), while recent neural survival models, such as DeepSurv (Katzman et al., 2018), DeepHit (Lee et al., 2018), and Dynamic-DeepHit (Lee et al., 2019), capture temporal dynamics but are restricted to tabular data, leaving multimodal integration underexplored. Furthermore, multimodal integration exploits complementary information across heterogeneous sources, imaging captures spatial phenotypes orthogonal to tabular measurements, enabling more expressive joint representations that reduce predictive uncertainty and mitigate unimodal information bottlenecks(Guarrasi et al., 2025).

To address the challenges of heterogeneity, irregular sampling, and censored event modeling in temporal multimodal survival analysis, we propose MultiTimeSurv, a principled and general framework designed to capture complex temporal dependencies across diverse clinical modalities. The core idea is to explicitly model non-stationary patterns in tabular data, sparsity in temporal sequences, and semantic alignment across modalities within a unified survival analysis framework. Specifically, MultiTimeSurv introduces three key components: (1) periodic and piecewise linear transformations to flexibly encode mixed categorical–continuous tabular features, enabling the capture of complex, non-stationary relationships; (2) temporal attention mechanisms that operate directly on irregularly sampled sequences, explicitly modeling sparsity and missingness; and (3) semantically-informed multimodal fusion via an extension of the CheXReport architecture (Zeiser et al., 2024), which jointly extracts and aligns visual features and textual semantics to produce robust cross-modal representations. MultiTimeSurv achieves state-of-the-art performance across two datasets, outperforming both general-purpose and survival-specific baselines. Ablation studies further confirm that each module contributes meaningfully, with multimodal fusion consistently improving performance across prediction horizons. Our contributions can be summarized as follows:

- We introduce MultiTimeSurv, a general-purpose survival analysis model that can process multimodal and longitudinal data by transforming the data into dense vector representations, allowing for the capture of multimodal information from censored and missing data over time.

- We introduce embedding techniques for categorical and continuous variables, transforming them into dense vector representations. This allows the capture of complex, non-linear relationships within the data, improving the model's ability to integrate and interpret diverse types of patient information for more accurate survival analysis.

- We employ a multitask learning approach to handle multiple competing health risks simultaneously. This enables the identification and differentiation of risk factors, providing a comprehensive assessment of patient prognosis and facilitating treatment strategies.

- We conduct an extensive experimental evaluation across two datasets (MDH and SYMILE-MIMIC), demonstrating that each architectural component contributes meaningfully and that MultiTimeSurv achieves state-of-the-art performance in both unimodal and multimodal survival prediction.

## 2 RELATED WORK

Early works such as DeepSurv (Katzman et al., 2018) extended the Cox proportional hazards framework by replacing linear predictors with neural networks, enabling the capture of complex non-linear covariate effects while preserving hazard ratio interpretability. However, these methods inherited the restrictive proportional hazards assumption and struggled with time-varying covariates. Discrete-time formulations marked a significant shift: DeepHit (Lee et al., 2018) introduced a multi-task learning framework to model the probability mass function of survival times directly, naturally accommodating competing risks. Extensions such as N-MTLR (Fotso, 2018) and PCHazard (Kvamme et al., 2019) further advanced hazard function parameterization but remained confined to static tabular inputs. To address temporal dynamics, Dynamic-DeepHit (Lee et al., 2019) incorporated recurrent neural networks with attention to model longitudinal covariates, achieving state-of-the-art performance but still limited by RNN weaknesses (e.g., long-range dependencies, irregular sampling). Recent approaches have shifted toward transformers, with continuous-time survival trans-

formers (Kvamme et al., 2019) leveraging self-attention for scalable temporal modeling. In parallel, multimodal survival models have emerged, such as Deep-CR MTLR (Kim et al., 2021), which integrates imaging and clinical features for competing risks, and SAMVAE (Garrido et al., 2025), which learns continuous-time multimodal survival distributions through variational inference. More recent methods, such as HySurvPred (Yang et al., 2025), employ hyperbolic embeddings and contrastive learning to capture censored and ordinal outcomes more effectively.

While recent advances have improved survival prediction, existing methods remain fragmented and fail to jointly address heterogeneous features, irregular sampling, and cross-modal alignment. Current approaches either handle temporal dynamics in isolation or focus on static multimodal fusion, leaving a gap for unified architectures that can effectively integrate both temporal and multimodal data. To overcome these limitations, we propose *MultiTimeSurv*, a principled framework that integrates multimodal, temporal, and tabular representations for survival analysis.

## 3 MULTITIMESURV: TEMPORAL MULTIMODAL NETWORKS FOR DYNAMIC SURVIVAL ANALYSIS

This section presents MultiTimeSurv, a general-purpose architecture for temporal multimodal survival analysis. In Figure 3, we present an overview of the MultiTimeSurv architecture. The framework jointly processes heterogeneous inputs, including clinical, laboratory, and imaging modalities, through modality-specific encoders and fusion layers that align representations in a shared latent space. By explicitly modeling temporal dynamics, irregular sampling, and cross-modal interactions, MultiTimeSurv captures a more comprehensive and clinically meaningful characterization of patient trajectories. This design enables more accurate survival predictions while providing interpretable representations of underlying risk factors, facilitating personalized treatment strategies and improved clinical decision support.

### 3.1 PROBLEM FORMULATION

We consider a longitudinal survival setting with multimodal inputs and competing risks. Let the dataset consist of $N$ subjects (e.g., a hospitalized patient). For each subject $i \in \{1, \ldots, N\}$ we observe:

$$X_i = (C_i, M_i, I_i), \quad Y_i = (\rho_i, \delta_i), \tag{1}$$

where $C_i$, $M_i$, and $I_i$ denote tabular covariates, missingness indicators, and imaging data, respectively. The survival outcome is represented by the event time $\rho_i \in \{1, \ldots, T\}$ and event type $\delta_i \in \{0, 1, \ldots, K\}$, with $\delta_i = 0$ indicating right-censoring.

**Multimodal longitudinal covariates.** - $C_i = \{c_{i,j,t}\}$ contains mixed categorical and continuous variables for covariate $j$ at time $t$. - $M_i = \{m_{i,j,t}\}$ is a binary mask indicating whether $c_{i,j,t}$ is missing. - $I_i = \{(\iota_{i,t}, \epsilon_{i,t})\}$ denotes imaging modalities (e.g., chest X-rays) paired with optional textual reports at irregularly sampled time points.

**Modeling objective.** The goal is to learn a mapping

$$f_\theta : X_i \mapsto P_i(t, k), \tag{2}$$

that outputs discrete hazard probabilities

$$P_i(t, k) = \Pr(\rho_i = t, \delta_i = k \mid X_i, \rho_i \geq t), \tag{3}$$

for each time $t \in \{1, \ldots, T\}$ and event type $k \in \{1, \ldots, K\}$. This formulation naturally accommodates competing risks and censored observations.

**Challenges.** This setup introduces several key challenges: (i) *heterogeneous covariates*, as $C_i$ combines categorical, continuous, and missing values; (ii) *irregular temporal sampling*, since not all modalities are observed at each time step; and (iii) *multimodal alignment*, as image–text pairs must be integrated with tabular signals for coherent survival prediction. Addressing these challenges requires an architecture that can flexibly embed heterogeneous features, handle missingness without heavy imputation, and fuse modalities while respecting temporal dynamics.

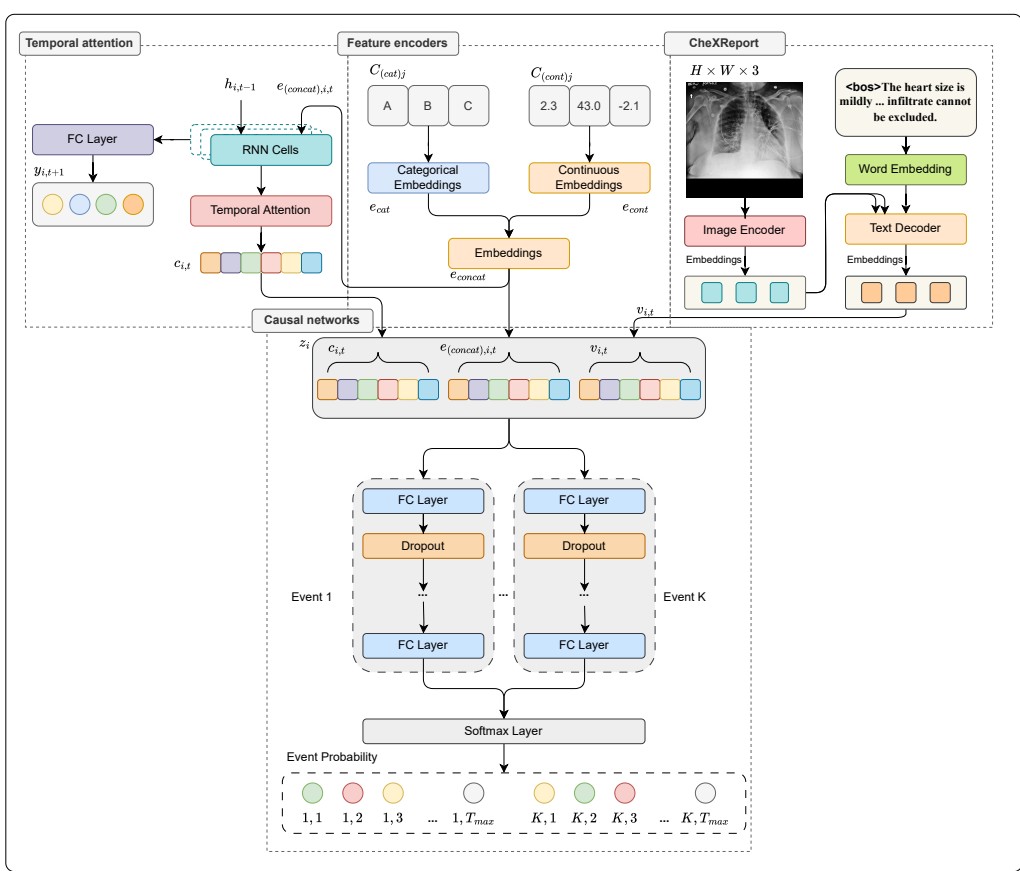

Figure 1: MultiTimeSurv model. Feature encoders process the tabular data, generating a vector of embeddings $e_{concat}$. Patient $i$ has the samples up to time $t$ processed in Temporal Attention, which generates a contextual vector $c_{i,t}$. The chest X-ray exam is processed by CheXReport, which produces a latent space vector $v_{i,t}$ with the main features identified. The vectors are concatenated $z_i$ and processed in the multitask networks to generate the probability for each time $t$ and event $k$.

## 3.2 FEATURE ENCODERS

**Categorical Embeddings.** For each categorical variable $j \in J_{cat}$ with vocabulary $\mathcal{V}_j$, we learn embeddings that incorporate frequency information:

$$e_{cat,j}(c_{i,j,t}) = E_j[\text{idx}(c_{i,j,t})] + f_j \cdot \log(\text{freq}(c_{i,j,t}) + 1) \tag{4}$$

where $E_j \in \mathbb{R}^{|\mathcal{V}_j| \times d}$ are learned embeddings and $f_j \in \mathbb{R}^d$ weights category frequency.

**Continuous Embeddings.** We combine three complementary strategies: periodic embeddings for cyclical patterns, piecewise embeddings for distribution modeling, and distributional embeddings for statistical properties:

$$e_{per,j}(c) = [\sin(2\pi W_{per,j}c + \phi_j), \cos(2\pi W_{per,j}c + \phi_j)] \tag{5}$$

$$e_{piece,j}(c) = \sum_{b=1}^{B} w_b \cdot \text{ReLU}\left(\frac{c - q_{j,b-1}}{q_{j,b} - q_{j,b-1}} - 1\right) \tag{6}$$

$$e_{dist,j}(c) = \text{MLP}_{dist}([\mu_j, \sigma_j, \text{skew}_j, \text{kurt}_j]) \tag{7}$$

The final continuous embedding fuses all representations:

$$e_{cont,j}(c) = W_{fusion,j}[e_{per,j}(c), e_{piece,j}(c), e_{dist,j}(c), c] + b_{fusion,j} \tag{8}$$

**Missing Data.** The dataset may be sparse and not have all covariates collected at all time points $t$. Therefore, for each covariate $j$ missing from patient $i$, we assign a value $-\infty$. To incorporate information into the MultiTimeSurv model regarding missing data, we provide the model with a mask $M = \{m_{i,1}, m_{i,2}, \ldots, m_{i,j}\}$, where: $m_{i,j} = 1$ if $x_{i,j} = -\infty$, $0$ otherwise

## 3.3 TEMPORAL ATTENTION MECHANISM

**Temporal Processing.** We employ a Gated Recurrent Unit (GRU) to encode the temporal sequence of patient embeddings. At each time step $t$, the GRU updates the hidden state using the current input $e_{(concat),i,t}$ and the previous state $h_{i,t-1}$:

$$h_{i,t} = GRU(e_{(concat),i,t}, h_{i,t-1}). \tag{9}$$

The sequence of hidden states $H_i = [h_{i,1}, \ldots, h_{i,T}] \in \mathbb{R}^{T \times d_{(hidden)}}$ is aggregated into a context vector $C_i$ via an attention mechanism:

$$C_i = \sum_{t=1}^{T} \varsigma_{i,t} h_{i,t}, \quad \varsigma_{i,t} = \frac{\exp(e_{i,t})}{\sum_{t=1}^{T} \exp(e_{i,t})}, \quad e_{i,t} = q^\top h_{i,t}, \tag{10}$$

where $q$ is a trainable query vector.

The context vector $C_i$ provides a compact summary of temporally relevant information, which is used to produce a longitudinal prediction one step ahead:

$$y_{i,t+1} = WC_i + b. \tag{11}$$

This formulation regularizes the temporal network while preserving predictive information (Lee et al., 2019).

## 3.4 CHEXREPORT

CheXReport adopts a fully transformer-based encoder–decoder architecture for joint visual–textual representation learning. Unlike CNN–RNN hybrids, our design leverages Swin Transformer blocks

in both encoder and decoder to capture hierarchical visual features from chest X-rays and align them with textual embeddings for report generation. The encoder applies stacked Swin Transformer blocks (Liu et al., 2021), which progressively merge patches to build multi-scale feature maps while reducing complexity compared to ViT (Dosovitskiy et al., 2021). Local attention within windows is implemented via $W–MSA$ and $SW–MSA$, where shifted partitions preserve cross-window context. Formally, self-attention is computed as

$$Attention(Q, K, V) = \text{softmax}\left(\frac{QK^T}{\sqrt{d}} + B\right) V, \tag{12}$$

with relative position bias $B \in \mathbb{R}^{M^2 \times M^2}$. Each block alternates $W–MSA$ and $SW–MSA$ with residual MLP layers and layer normalization, ensuring both local efficiency and global consistency.

The decoder receives the input report sequence, tokenized into $K$ units and embedded with multi-lingual BERT vectors. After adding positional embeddings, tokens are processed through masked self-attention, layer normalization, and cross-attention to integrate image-derived features. Finally, a linear projection maps the hidden states to the vocabulary space, generating the report suggestion. This design enables CheXReport to extract clinically relevant visual features, align them with textual semantics, and produce context-aware radiology reports.

## 3.5 MULTITASK NETWORKS

We adopt a multitask learning structure to estimate the specific risk $k$ for each patient $i$ at time $t$. Each network specializes in one event type while sharing intermediate representations, allowing MultiTimeSurv to capture dependencies across risks without interfering with event-specific patterns. The input to each multitask network combines three components: the temporal context vector $c_{i,t}$, the most recent embedding $e_{(concat),i,t}$, and the CheXReport latent vector $v_{i,t}$, formally:

$$z_i = \left[c_{i,t}, e_{(concat),i,t}, v_{i,t}\right]. \tag{13}$$

Each multitask network is a feed-forward model with $L$ hidden layers. The first hidden layer is:

$$h^{(0)} = ReLU(W^{(0)}z_i + b^{(0)}), \tag{14}$$

followed by

$$h^{(l)} = ReLU(W^{(l)}h^{(l-1)} + b^{(l)}), \quad l = 1, \ldots, L, \tag{15}$$

with dropout applied at each layer:

$$h^{(l)} = Dropout(h^{(l)}, p). \tag{16}$$

The output layer for risk $k$ is given by:

$$o_k = W^{(L+1)}h^{(L)} + b^{(L+1)}. \tag{17}$$

Outputs across all risks are concatenated:

$$O = [o_1, o_2, \ldots, o_K], \tag{18}$$

and normalized with a softmax to yield event probabilities over time:

$$P = softmax(O). \tag{19}$$

## 3.6 MULTITIMESURV MODEL OPTIMIZATION

MultiTimeSurv is trained with a composite loss

$$L_{(total)} = L_1 + L_2 + L_3, \tag{20}$$

where $L_1$ captures survival events via binary cross-entropy, $L_2$ enforces temporal consistency with a masked MSE that ignores missing values (Lee et al., 2019), and $L_3$ regularizes CheXReport through cross-entropy with stochastic attention (Xu et al., 2015). Full definitions of these loss terms are provided in Appendix B.

The computational complexity is $O(HW \cdot d_v + T^2 \cdot d + K \cdot L \cdot d^2)$ with space requirements $O(HW + T \cdot |J| \cdot d + N_v \cdot d_v)$, remaining tractable for typical clinical sequences with $T < 100$.

## 4 SURVIVAL ANALYSIS RESULTS

We evaluate MultiTimeSurv on two datasets of increasing complexity: (i) MDH, a multimodal hospital dataset with longitudinal clinical, laboratory, and imaging data; and (ii) SYMILE-MIMIC, a multimodal dataset integrating EHR and imaging. We report C-index and Brier score across prediction times $t \in \{1, 3, 5, 7\}$ and evaluation horizons $\Delta t \in \{1, 3, 5, 7\}$].

### 4.1 RESULTS ON MDH

Table 1 presents the results on the multimodal MDH dataset. Across all prediction horizons, MultiTimeSurv achieves the best discriminative performance, with gains of up to $+0.300$ in C-index compared to CoxPH and $+0.100$ over Dynamic-DeepHit. In terms of calibration, MultiTimeSurv remains competitive, with Brier scores that are consistently among the lowest across both short- and long-term horizons. Importantly, MultiTimeSurv maintains stable performance even at the most challenging setting ($t = 7, \Delta t = 7$), where most baselines show substantial degradation.

Table 1: Comparison of MultiTimeSurv with various methods across prediction times $t$ and horizons $\Delta t$. Left: C-index (higher is better). Right: Brier score (lower is better). Values are mean $\pm$ std.

| | | C-index | | | | Brier Score | | | |
|---|---|---|---|---|---|---|---|---|---|
| $t$ | Algorithms | $\Delta t = 1$ | $\Delta t = 3$ | $\Delta t = 5$ | $\Delta t = 7$ | $\Delta t = 1$ | $\Delta t = 3$ | $\Delta t = 5$ | $\Delta t = 7$ |
| | CoxPH[†] | $0.377 \pm 0.08$ | $0.312 \pm 0.01$ | $0.380 \pm 0.01$ | $0.406 \pm 0.04$ | $0.010 \pm 0.00$ | $0.023 \pm 0.00$ | $0.037 \pm 0.00$ | $0.056 \pm 0.00$ |
| | CoxCC[†] | $0.463 \pm 0.07$ | $0.340 \pm 0.01$ | $0.400 \pm 0.01$ | $0.436 \pm 0.02$ | $0.010 \pm 0.00$ | $0.023 \pm 0.00$ | $0.038 \pm 0.00$ | $0.057 \pm 0.00$ |
| | DeepSurv[†] | $0.361 \pm 0.01$ | $0.326 \pm 0.07$ | $0.378 \pm 0.09$ | $0.410 \pm 0.02$ | $0.010 \pm 0.00$ | $0.023 \pm 0.00$ | $0.037 \pm 0.00$ | $0.056 \pm 0.00$ |
| | PCHazard[†] | $0.555 \pm 0.08$ | $0.552 \pm 0.01$ | $0.506 \pm 0.03$ | $0.524 \pm 0.03$ | $0.028 \pm 0.07$ | $0.054 \pm 0.05$ | $0.066 \pm 0.04$ | $0.071 \pm 0.03$ |
| $t = 1$ | DeepHit[†] | $0.533 \pm 0.01$ | $0.462 \pm 0.01$ | $0.480 \pm 0.06$ | $0.456 \pm 0.07$ | $0.012 \pm 0.00$ | $0.026 \pm 0.00$ | $0.041 \pm 0.00$ | $0.061 \pm 0.00$ |
| | N-MTLR[†] | $0.453 \pm 0.01$ | $0.452 \pm 0.06$ | $0.465 \pm 0.03$ | $0.453 \pm 0.05$ | $0.011 \pm 0.00$ | $0.026 \pm 0.00$ | $0.043 \pm 0.00$ | $0.066 \pm 0.00$ |
| | DynamicDeepHit[†] | $0.666 \pm 0.02$ | $0.622 \pm 0.02$ | $0.629 \pm 0.01$ | $0.648 \pm 0.01$ | $0.060 \pm 0.00$ | $0.092 \pm 0.00$ | $0.130 \pm 0.00$ | $0.163 \pm 0.00$ |
| | Model A [†] | $0.693 \pm 0.02$ | $0.708 \pm 0.02$ | $0.702 \pm 0.02$ | $0.701 \pm 0.01$ | $0.066 \pm 0.00$ | $0.074 \pm 0.00$ | $0.102 \pm 0.00$ | $0.147 \pm 0.00$ |
| | Model B | $0.695 \pm 0.01$ | $0.711 \pm 0.02$ | $0.701 \pm 0.01$ | $0.703 \pm 0.01$ | $0.065 \pm 0.00$ | $0.073 \pm 0.00$ | $0.103 \pm 0.00$ | $0.146 \pm 0.00$ |
| | **MultiTimeSurv** | $\mathbf{0.723 \pm 0.08}$ | $\mathbf{0.735 \pm 0.01}$ | $\mathbf{0.711 \pm 0.02}$ | $\mathbf{0.706 \pm 0.01}$ | $\mathbf{0.071 \pm 0.00}$ | $\mathbf{0.024 \pm 0.01}$ | $\mathbf{0.097 \pm 0.00}$ | $\mathbf{0.147 \pm 0.00}$ |
| | CoxPH[†] | $0.282 \pm 0.07$ | $0.388 \pm 0.09$ | $0.418 \pm 0.07$ | $0.420 \pm 0.03$ | $0.036 \pm 0.00$ | $0.051 \pm 0.00$ | $0.072 \pm 0.00$ | $0.098 \pm 0.00$ |
| | CoxCC[†] | $0.362 \pm 0.01$ | $0.435 \pm 0.01$ | $0.461 \pm 0.02$ | $0.463 \pm 0.05$ | $0.037 \pm 0.00$ | $0.052 \pm 0.00$ | $0.073 \pm 0.00$ | $0.073 \pm 0.00$ |
| | DeepSurv[†] | $0.340 \pm 0.01$ | $0.404 \pm 0.01$ | $0.430 \pm 0.02$ | $0.439 \pm 0.04$ | $0.036 \pm 0.00$ | $0.051 \pm 0.00$ | $0.072 \pm 0.00$ | $0.097 \pm 0.00$ |
| | PCHazard[†] | $0.541 \pm 0.04$ | $0.492 \pm 0.03$ | $0.503 \pm 0.03$ | $0.476 \pm 0.04$ | $0.080 \pm 0.02$ | $0.082 \pm 0.01$ | $0.081 \pm 0.01$ | $0.107 \pm 0.00$ |
| $t = 3$ | DeepHit[†] | $0.472 \pm 0.01$ | $0.534 \pm 0.06$ | $0.510 \pm 0.06$ | $0.495 \pm 0.06$ | $0.040 \pm 0.00$ | $0.056 \pm 0.00$ | $0.077 \pm 0.00$ | $0.104 \pm 0.01$ |
| | N-MTLR[†] | $0.410 \pm 0.07$ | $0.452 \pm 0.05$ | $0.447 \pm 0.05$ | $0.460 \pm 0.04$ | $0.042 \pm 0.00$ | $0.060 \pm 0.00$ | $0.085 \pm 0.00$ | $0.117 \pm 0.01$ |
| | DynamicDeepHit[†] | $0.584 \pm 0.04$ | $0.635 \pm 0.03$ | $0.639 \pm 0.02$ | $0.633 \pm 0.02$ | $0.097 \pm 0.00$ | $0.136 \pm 0.00$ | $0.168 \pm 0.00$ | $0.186 \pm 0.00$ |
| | Model A [†] | $0.740 \pm 0.01$ | $0.720 \pm 0.02$ | $0.728 \pm 0.02$ | $0.714 \pm 0.02$ | $0.094 \pm 0.00$ | $0.141 \pm 0.00$ | $0.172 \pm 0.00$ | $0.191 \pm 0.00$ |
| | Model B | $0.742 \pm 0.01$ | $0.718 \pm 0.01$ | $0.730 \pm 0.01$ | $0.715 \pm 0.01$ | $0.094 \pm 0.00$ | $0.142 \pm 0.00$ | $0.171 \pm 0.00$ | $0.190 \pm 0.00$ |
| | **MultiTimeSurv** | $\mathbf{0.742 \pm 0.00}$ | $\mathbf{0.729 \pm 0.03}$ | $\mathbf{0.735 \pm 0.00}$ | $\mathbf{0.726 \pm 0.00}$ | $\mathbf{0.088 \pm 0.01}$ | $\mathbf{0.112 \pm 0.00}$ | $\mathbf{0.152 \pm 0.00}$ | $\mathbf{0.180 \pm 0.00}$ |
| | CoxPH[†] | $0.387 \pm 0.09$ | $0.420 \pm 0.01$ | $0.423 \pm 0.02$ | $0.396 \pm 0.02$ | $0.068 \pm 0.01$ | $0.092 \pm 0.00$ | $0.119 \pm 0.00$ | $0.139 \pm 0.01$ |
| | CoxCC[†] | $0.359 \pm 0.01$ | $0.438 \pm 0.04$ | $0.451 \pm 0.06$ | $0.441 \pm 0.04$ | $0.069 \pm 0.01$ | $0.094 \pm 0.00$ | $0.120 \pm 0.00$ | $0.140 \pm 0.00$ |
| | DeepSurv[†] | $0.362 \pm 0.08$ | $0.425 \pm 0.05$ | $0.437 \pm 0.03$ | $0.438 \pm 0.03$ | $0.068 \pm 0.01$ | $0.092 \pm 0.00$ | $0.118 \pm 0.00$ | $0.137 \pm 0.00$ |
| | PCHazard[†] | $0.454 \pm 0.04$ | $0.487 \pm 0.05$ | $0.480 \pm 0.04$ | $0.486 \pm 0.04$ | $0.087 \pm 0.01$ | $0.099 \pm 0.01$ | $0.136 \pm 0.01$ | $0.151 \pm 0.01$ |
| $t = 5$ | DeepHit[†] | $0.547 \pm 0.04$ | $0.526 \pm 0.03$ | $0.512 \pm 0.06$ | $0.523 \pm 0.03$ | $0.073 \pm 0.01$ | $0.098 \pm 0.01$ | $0.126 \pm 0.01$ | $0.149 \pm 0.01$ |
| | N-MTLR[†] | $0.457 \pm 0.01$ | $0.451 \pm 0.06$ | $0.467 \pm 0.04$ | $0.475 \pm 0.04$ | $0.080 \pm 0.00$ | $0.109 \pm 0.01$ | $0.144 \pm 0.01$ | $0.174 \pm 0.01$ |
| | DynamicDeepHit[†] | $0.605 \pm 0.02$ | $0.617 \pm 0.03$ | $0.614 \pm 0.02$ | $0.611 \pm 0.02$ | $0.143 \pm 0.00$ | $0.174 \pm 0.00$ | $0.192 \pm 0.00$ | $0.195 \pm 0.00$ |
| | Model A [†] | $0.722 \pm 0.02$ | $0.728 \pm 0.01$ | $0.720 \pm 0.01$ | $0.706 \pm 0.02$ | $0.134 \pm 0.00$ | $0.194 \pm 0.00$ | $0.190 \pm 0.00$ | $0.203 \pm 0.00$ |
| | Model B | $0.725 \pm 0.01$ | $0.729 \pm 0.01$ | $0.721 \pm 0.01$ | $0.709 \pm 0.01$ | $0.133 \pm 0.00$ | $0.193 \pm 0.00$ | $0.189 \pm 0.00$ | $0.202 \pm 0.00$ |
| | **MultiTimeSurv** | $\mathbf{0.726 \pm 0.06}$ | $\mathbf{0.731 \pm 0.01}$ | $\mathbf{0.722 \pm 0.03}$ | $\mathbf{0.714 \pm 0.04}$ | $\mathbf{0.117 \pm 0.00}$ | $\mathbf{0.161 \pm 0.01}$ | $\mathbf{0.182 \pm 0.01}$ | $\mathbf{0.199 \pm 0.01}$ |
| | CoxPH[†] | $0.371 \pm 0.08$ | $0.404 \pm 0.05$ | $0.382 \pm 0.03$ | $0.358 \pm 0.02$ | $0.117 \pm 0.01$ | $0.146 \pm 0.01$ | $0.163 \pm 0.01$ | $0.177 \pm 0.01$ |
| | CoxCC[†] | $0.420 \pm 0.06$ | $0.443 \pm 0.07$ | $0.437 \pm 0.04$ | $0.430 \pm 0.03$ | $0.119 \pm 0.01$ | $0.147 \pm 0.01$ | $0.164 \pm 0.01$ | $0.178 \pm 0.00$ |
| | DeepSurv[†] | $0.396 \pm 0.06$ | $0.432 \pm 0.06$ | $0.440 \pm 0.04$ | $0.422 \pm 0.03$ | $0.117 \pm 0.01$ | $0.144 \pm 0.01$ | $0.161 \pm 0.00$ | $0.175 \pm 0.00$ |
| | PCHazard[†] | $0.494 \pm 0.09$ | $0.478 \pm 0.09$ | $0.496 \pm 0.08$ | $0.479 \pm 0.08$ | $0.133 \pm 0.01$ | $0.161 \pm 0.01$ | $0.180 \pm 0.02$ | $0.203 \pm 0.01$ |
| $t = 7$ | DeepHit[†] | $0.563 \pm 0.01$ | $0.510 \pm 0.07$ | $0.520 \pm 0.04$ | $0.533 \pm 0.07$ | $0.124 \pm 0.01$ | $0.154 \pm 0.01$ | $0.175 \pm 0.01$ | $0.191 \pm 0.01$ |
| | N-MTLR[†] | $0.420 \pm 0.01$ | $0.467 \pm 0.05$ | $0.476 \pm 0.04$ | $0.452 \pm 0.03$ | $0.140 \pm 0.01$ | $0.178 \pm 0.02$ | $0.206 \pm 0.02$ | $0.231 \pm 0.02$ |
| | DynamicDeepHit[†] | $0.605 \pm 0.02$ | $0.617 \pm 0.03$ | $0.614 \pm 0.02$ | $0.611 \pm 0.02$ | $0.185 \pm 0.00$ | $0.202 \pm 0.00$ | $0.202 \pm 0.00$ | $0.212 \pm 0.00$ |
| | Model A [†] | $0.694 \pm 0.03$ | $0.696 \pm 0.02$ | $0.695 \pm 0.01$ | $0.688 \pm 0.01$ | $0.165 \pm 0.00$ | $0.200 \pm 0.00$ | $0.208 \pm 0.00$ | $0.211 \pm 0.00$ |
| | Model B | $0.692 \pm 0.02$ | $0.698 \pm 0.01$ | $0.696 \pm 0.01$ | $0.690 \pm 0.01$ | $0.164 \pm 0.00$ | $0.199 \pm 0.00$ | $0.206 \pm 0.00$ | $0.209 \pm 0.00$ |
| | **MultiTimeSurv** | $\mathbf{0.725 \pm 0.01}$ | $\mathbf{0.715 \pm 0.02}$ | $\mathbf{0.702 \pm 0.03}$ | $\mathbf{0.695 \pm 0.03}$ | $\mathbf{0.150 \pm 0.00}$ | $\mathbf{0.168 \pm 0.00}$ | $\mathbf{0.186 \pm 0.00}$ | $\mathbf{0.193 \pm 0.01}$ |

[†] Trained only with tabular data.

Model A: baseline with an embedding model.

Model B: baseline with an embedding model and ResNet50v2 as image encoder.

Across prediction times, MultiTimeSurv consistently achieves the highest C-index values compared to all baselines, showing stronger discriminative ability. Traditional models such as CoxTime and DeepSurv tend to underperform, particularly at longer horizons. For Brier scores, simple tabular-only methods yield the lowest values at very short horizons (reflecting good short-term calibration), but MultiTimeSurv remains competitive and achieves better overall balance between discrimination and calibration. Among the baselines, Model B and Model C perform competitively, but the inclusion of multimodal features in MultiTimeSurv further improves performance, highlighting the benefit of joint representation learning.

## 4.2 RESULTS ON SYMILE-MIMIC

The SYMILE-MIMIC dataset combines structured EHR data with longitudinal survival outcomes. Results in Table 2 demonstrate that MultiTimeSurv achieves consistently higher concordance indices compared to classical baselines such as CoxPH and CoxCC, as well as neural methods including DeepHit and MTLR. In particular, gains are most pronounced at longer horizons ($\Delta t = 5, 7$), where standard models tend to degrade.

Table 2: Comparison of MultiTimeSurv with various methods on SYMILE-MIMIC across prediction times $t$ and horizons $\Delta t$. Left: C-index (higher is better). Right: Brier score (lower is better). Values are mean $\pm$ std.

| $t$ | Algorithms | C-index | | | | Brier Score | | | |
|---|---|---|---|---|---|---|---|---|---|
| | | $\Delta t = 1$ | $\Delta t = 3$ | $\Delta t = 5$ | $\Delta t = 7$ | $\Delta t = 1$ | $\Delta t = 3$ | $\Delta t = 5$ | $\Delta t = 7$ |
| $t=1$ | CoxPH[†] | $0.150 \pm 0.08$ | $0.205 \pm 0.06$ | $0.217 \pm 0.04$ | $0.234 \pm 0.03$ | $0.003 \pm 0.00$ | $0.013 \pm 0.00$ | $0.024 \pm 0.00$ | $0.036 \pm 0.00$ |
| | CoxCC[†] | $0.381 \pm 0.20$ | $0.407 \pm 0.15$ | $0.401 \pm 0.13$ | $0.416 \pm 0.13$ | $0.003 \pm 0.00$ | $0.013 \pm 0.00$ | $0.025 \pm 0.00$ | $0.038 \pm 0.00$ |
| | PCHazard[†] | $0.471 \pm 0.08$ | $0.482 \pm 0.13$ | $0.475 \pm 0.05$ | $0.476 \pm 0.03$ | $0.343 \pm 0.15$ | $0.584 \pm 0.07$ | $0.695 \pm 0.04$ | $0.740 \pm 0.03$ |
| | PMF[†] | $0.502 \pm 0.08$ | $0.479 \pm 0.10$ | $0.492 \pm 0.07$ | $0.484 \pm 0.07$ | $0.011 \pm 0.00$ | $0.025 \pm 0.00$ | $0.040 \pm 0.01$ | $0.060 \pm 0.01$ |
| | DeepHit[†] | $0.642 \pm 0.10$ | $0.602 \pm 0.04$ | $0.579 \pm 0.04$ | $0.573 \pm 0.03$ | $0.003 \pm 0.00$ | $0.013 \pm 0.00$ | $0.025 \pm 0.00$ | $0.038 \pm 0.00$ |
| | MTLR[†] | $0.481 \pm 0.11$ | $0.498 \pm 0.07$ | $0.505 \pm 0.09$ | $0.513 \pm 0.09$ | $0.027 \pm 0.05$ | $0.037 \pm 0.05$ | $0.050 \pm 0.05$ | $0.065 \pm 0.05$ |
| | **MultiTimeSurv** | $\mathbf{0.682 \pm 0.01}$ | $\mathbf{0.711 \pm 0.02}$ | $\mathbf{0.674 \pm 0.02}$ | $\mathbf{0.668 \pm 0.02}$ | $0.226 \pm 0.15$ | $0.245 \pm 0.13$ | $0.247 \pm 0.13$ | $0.245 \pm 0.12$ |
| $t=3$ | CoxPH[†] | $0.260 \pm 0.08$ | $0.237 \pm 0.03$ | $0.252 \pm 0.03$ | $0.261 \pm 0.02$ | $0.023 \pm 0.00$ | $0.035 \pm 0.01$ | $0.047 \pm 0.01$ | $0.058 \pm 0.01$ |
| | CoxCC[†] | $0.409 \pm 0.15$ | $0.415 \pm 0.13$ | $0.435 \pm 0.12$ | $0.449 \pm 0.10$ | $0.024 \pm 0.00$ | $0.037 \pm 0.01$ | $0.050 \pm 0.01$ | $0.063 \pm 0.01$ |
| | PCHazard[†] | $0.532 \pm 0.10$ | $0.488 \pm 0.05$ | $0.491 \pm 0.05$ | $0.489 \pm 0.02$ | $0.795 \pm 0.03$ | $0.841 \pm 0.01$ | $0.849 \pm 0.01$ | $0.830 \pm 0.01$ |
| | PMF[†] | $0.474 \pm 0.15$ | $0.501 \pm 0.07$ | $0.480 \pm 0.11$ | $0.510 \pm 0.05$ | $0.039 \pm 0.01$ | $0.055 \pm 0.01$ | $0.077 \pm 0.01$ | $0.103 \pm 0.01$ |
| | DeepHit[†] | $0.543 \pm 0.10$ | $0.548 \pm 0.08$ | $0.549 \pm 0.07$ | $0.547 \pm 0.07$ | $0.024 \pm 0.00$ | $0.037 \pm 0.01$ | $0.050 \pm 0.01$ | $0.063 \pm 0.01$ |
| | MTLR[†] | $0.504 \pm 0.10$ | $0.514 \pm 0.11$ | $0.525 \pm 0.10$ | $0.517 \pm 0.08$ | $0.049 \pm 0.05$ | $0.062 \pm 0.05$ | $0.079 \pm 0.05$ | $0.095 \pm 0.06$ |
| | **MultiTimeSurv** | $\mathbf{0.744 \pm 0.02}$ | $\mathbf{0.736 \pm 0.01}$ | $\mathbf{0.737 \pm 0.01}$ | $\mathbf{0.726 \pm 0.01}$ | $0.161 \pm 0.13$ | $0.200 \pm 0.14$ | $0.239 \pm 0.14$ | $0.234 \pm 0.14$ |
| $t=5$ | CoxPH[†] | $0.213 \pm 0.06$ | $0.265 \pm 0.06$ | $0.278 \pm 0.04$ | $0.287 \pm 0.04$ | $0.047 \pm 0.01$ | $0.058 \pm 0.01$ | $0.070 \pm 0.01$ | $0.080 \pm 0.01$ |
| | CoxCC[†] | $0.460 \pm 0.14$ | $0.464 \pm 0.11$ | $0.474 \pm 0.09$ | $0.472 \pm 0.08$ | $0.050 \pm 0.01$ | $0.063 \pm 0.01$ | $0.076 \pm 0.01$ | $0.088 \pm 0.01$ |
| | PCHazard[†] | $0.465 \pm 0.08$ | $0.493 \pm 0.08$ | $0.483 \pm 0.04$ | $0.501 \pm 0.04$ | $0.875 \pm 0.01$ | $0.866 \pm 0.01$ | $0.834 \pm 0.01$ | $0.801 \pm 0.02$ |
| | PMF[†] | $0.565 \pm 0.07$ | $0.478 \pm 0.11$ | $0.516 \pm 0.05$ | $0.514 \pm 0.03$ | $0.072 \pm 0.01$ | $0.097 \pm 0.01$ | $0.126 \pm 0.01$ | $0.149 \pm 0.01$ |
| | DeepHit[†] | $0.565 \pm 0.11$ | $0.544 \pm 0.06$ | $0.534 \pm 0.05$ | $0.529 \pm 0.05$ | $0.050 \pm 0.01$ | $0.063 \pm 0.01$ | $0.076 \pm 0.01$ | $0.088 \pm 0.01$ |
| | MTLR[†] | $0.555 \pm 0.11$ | $0.531 \pm 0.07$ | $0.513 \pm 0.05$ | $0.510 \pm 0.05$ | $0.076 \pm 0.05$ | $0.094 \pm 0.06$ | $0.111 \pm 0.06$ | $0.128 \pm 0.07$ |
| | **MultiTimeSurv** | $\mathbf{0.720 \pm 0.02}$ | $\mathbf{0.737 \pm 0.02}$ | $\mathbf{0.729 \pm 0.01}$ | $\mathbf{0.727 \pm 0.01}$ | $0.115 \pm 0.11$ | $0.157 \pm 0.12$ | $0.200 \pm 0.13$ | $0.223 \pm 0.13$ |
| $t=7$ | CoxPH[†] | $0.319 \pm 0.16$ | $0.297 \pm 0.10$ | $0.308 \pm 0.07$ | $0.314 \pm 0.06$ | $0.071 \pm 0.01$ | $0.081 \pm 0.01$ | $0.091 \pm 0.01$ | $0.102 \pm 0.01$ |
| | CoxCC[†] | $0.480 \pm 0.14$ | $0.488 \pm 0.08$ | $0.474 \pm 0.07$ | $0.469 \pm 0.06$ | $0.077 \pm 0.01$ | $0.089 \pm 0.01$ | $0.101 \pm 0.01$ | $0.114 \pm 0.02$ |
| | PCHazard[†] | $0.475 \pm 0.13$ | $0.479 \pm 0.09$ | $0.512 \pm 0.08$ | $0.500 \pm 0.06$ | $0.847 \pm 0.02$ | $0.805 \pm 0.02$ | $0.772 \pm 0.02$ | $0.738 \pm 0.02$ |
| | PMF[†] | $0.544 \pm 0.09$ | $0.560 \pm 0.05$ | $0.526 \pm 0.05$ | $0.515 \pm 0.05$ | $0.124 \pm 0.01$ | $0.154 \pm 0.01$ | $0.175 \pm 0.01$ | $0.193 \pm 0.01$ |
| | DeepHit[†] | $0.562 \pm 0.11$ | $0.536 \pm 0.08$ | $0.526 \pm 0.06$ | $0.527 \pm 0.06$ | $0.076 \pm 0.01$ | $0.089 \pm 0.01$ | $0.101 \pm 0.01$ | $0.113 \pm 0.01$ |
| | MTLR[†] | $0.523 \pm 0.12$ | $0.497 \pm 0.06$ | $0.494 \pm 0.06$ | $0.499 \pm 0.05$ | $0.109 \pm 0.06$ | $0.128 \pm 0.07$ | $0.144 \pm 0.07$ | $0.158 \pm 0.07$ |
| | **MultiTimeSurv** | $\mathbf{0.719 \pm 0.01}$ | $\mathbf{0.716 \pm 0.02}$ | $\mathbf{0.704 \pm 0.01}$ | $\mathbf{0.699 \pm 0.01}$ | $0.150 \pm 0.11$ | $0.169 \pm 0.12$ | $0.188 \pm 0.12$ | $0.197 \pm 0.12$ |

[†] Trained only with tabular data. MultiTimeSurv combines tabular and temporal information with multi-event modeling.

The results on the SYMILE-MIMIC dataset highlight consistent improvements of MultiTimeSurv over traditional tabular survival methods (CoxPH, CoxCC) and baselines, including PCHazard, PMF, DeepHit, and MTLR. Across all prediction times $t$ and horizons $\Delta t$, MultiTimeSurv achieves higher concordance indices, often exceeding $0.70$, indicating superior discriminative ability. While some neural baselines (e.g., DeepHit, PMF) approach competitive performance, their gains are less stable across horizons, and methods like PCHazard suffer from inflated Brier scores, reflecting poor calibration despite moderate ranking power. Importantly, MultiTimeSurv maintains balanced performance between discrimination (C-index) and calibration (Brier score), with particularly strong gains at longer horizons ($\Delta t = 5$ and 7). This suggests the model's ability to leverage both tabular covariates and temporal structure to capture long-term survival dynamics.

## 4.3 CASE STUDY ON MDH DATASET

To better understand the behavior of MultiTimeSurv, we present a qualitative analysis of patient-specific risk trajectories in the MDH test set (Figure 2). For each patient, the model outputs time-dependent risks for discharge and death, allowing us to assess whether predictions align with observed clinical outcomes.

**Patient A.** The predicted discharge risk increases as hospitalization progresses, peaking near the actual discharge date. This aligns with the observed event and suggests that MultiTimeSurv can capture gradual improvements in health status. Notably, death risk remains high in the early days, consistent with the severity of the overall cohort.

**Patient B.** This patient was right-censored (e.g., transferred or lost to follow-up). During the observed period, the model consistently assigned higher discharge risk relative to death, which may reflect a plausible transfer to a less critical facility.

Figure 2: Predicted independent risks for discharge (purple) and death (red) over a 50-day horizon. Stars mark the observed event type and time.

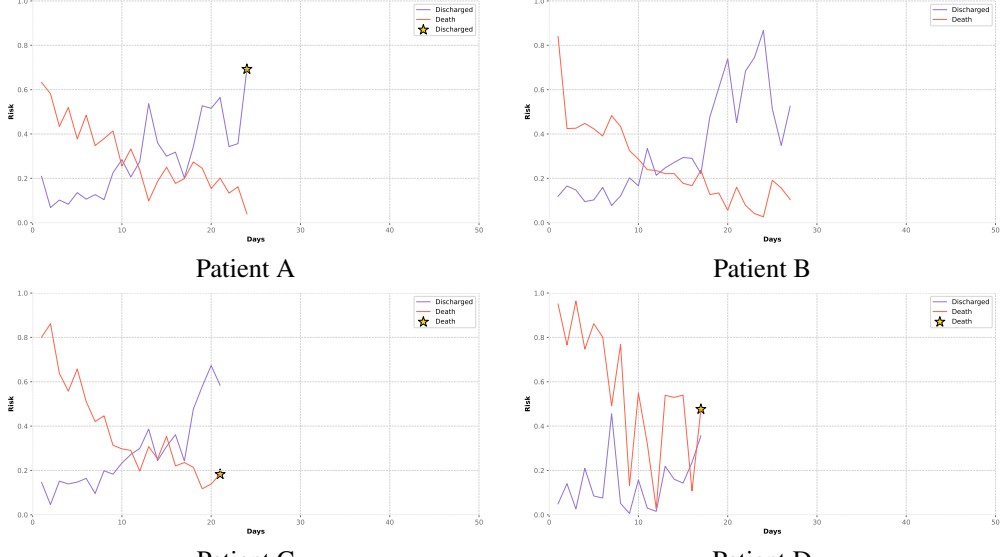

Patient A  Patient B

Patient C  Patient D

**Patient C.** In this case, the model did not assign the highest risk to the observed event. Early predictions emphasized death risk, which later decreased, while discharge risk increased but without clear alignment to the actual outcome. The intersection of risk trajectories highlights the model's uncertainty and evolving perception of patient status.

**Patient D.** Predictions for this patient exhibit fluctuations in death risk, potentially reflecting variable clinical stability during hospitalization. These oscillations suggest sensitivity of the model to dynamic health conditions and reinforce the importance of continuous monitoring for timely interventions.

## 5 CONCLUSION

In this paper, we presented MultiTimeSurv, a unified framework for multimodal temporal survival analysis that integrates clinical, laboratory, and imaging data. Our design combines periodic and piecewise embeddings for heterogeneous tabular variables, temporal attention for irregular sequences, and semantically aligned multimodal fusion via CheXReport. Evaluations on MDH and SYMILE-MIMIC highlight three key insights: (i) multimodal fusion consistently enhances discrimination and calibration, particularly in settings with irregular sampling; (ii) ablations confirm complementary contributions from each component, with temporal attention improving long-horizon stability and tabular embeddings strengthening short-term estimation; and (iii) the model generalizes well to unimodal benchmarks, outperforming classical and deep learning baselines. Limitations and directions for future research are discussed in Appendix F.

## ETHICS STATEMENT

This research was conducted in accordance with the Helsinki Declaration and received approval from the institutional ethics committee number [hidden]. All datasets contain de-identified patient information accessed through appropriate institutional agreements.

## REPRODUCIBILITY STATEMENT

Complete implementation is available at `https://github.com/multitimesurv/iclr2026`.

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

## A  SURVIVAL ANALYSIS

Survival analysis, or time-to-event analysis, studies the expected duration until one or more events occur (Kaplan & Meier, 1958). Applications range from time-to-death and cancer recurrence to mechanical component failure (Bradburn et al., 2003). The primary goals are to estimate the distribution of survival times, compare groups, and model the relationship between time-to-event and covariates (Cox, 1972; Bradburn et al., 2003).

### A.1  KEY FUNCTIONS

Survival data are typically described by the survival function $S(t)$ and the hazard function $\lambda(t)$ (Clark et al., 2003). The survival function represents the probability of surviving beyond time $t$:

$$S(t) = P(T > t), \tag{21}$$

where $T$ is the random variable denoting time-to-event. It is non-increasing, with $S(0) = 1$ and $\lim_{t \to \infty} S(t) = 0$ (Wang et al., 2019).

The hazard function, $\lambda(t)$, captures the instantaneous event rate at time $t$, conditional on survival up to $t$:

$$\lambda(t) = \lim_{\Delta t \to 0} \frac{P(t \le T < t + \Delta t \mid T \ge t)}{\Delta t}. \tag{22}$$

It is a rate rather than a probability, useful for identifying high- or low-risk periods (Clark et al., 2003; Wang et al., 2019).

### A.2  CENSORING

A key challenge in survival analysis is censoring, which arises when the event time is not fully observed (Kleinbaum et al., 2012). Common cases include patients not experiencing the event during follow-up, loss to follow-up, or occurrence of a competing event (Clark et al., 2003). Improperly handling censored data can bias estimates and distort conclusions (Lee et al., 2019). Moreover, high censoring reduces effective sample size, widening confidence intervals and lowering precision (Klein, 2003).

### A.3  SURVIVAL MODELS

The Kaplan–Meier estimator and the Cox proportional hazards (CoxPH) model are the most widely used survival models (Wang et al., 2019). The Kaplan–Meier estimator provides a non-parametric estimate of $S(t)$:

$$\hat{S}(t) = \prod_{t_i \le t} \left( 1 - \frac{d_i}{n_i} \right), \tag{23}$$

where $d_i$ is the number of events at $t_i$ and $n_i$ the number at risk just prior to $t_i$. While robust to censoring, it does not incorporate covariates or heterogeneity among subjects (Klein, 2003; Clark et al., 2003).

The CoxPH model, by contrast, estimates the hazard function conditioned on covariates $X = (X_1, \ldots, X_p)$ (Cox, 1972):

$$\lambda(t \mid X) = \lambda_0(t) \exp(\beta_1 X_1 + \cdots + \beta_p X_p), \tag{24}$$

where $\lambda_0(t)$ is the baseline hazard and $\beta$ are regression coefficients (Bradburn et al., 2003). The proportional hazards assumption—that hazard ratios remain constant over time—is central to CoxPH (Wang et al., 2019). Violations of this assumption may lead to biased or misleading results (Lee et al., 2019).

## B  LOSS FUNCTION DETAILS

For completeness, we provide the explicit formulations of the loss terms used to train Multi-TimeSurv.

## B.1 Survival Event Loss $L_1$

We adapt binary cross-entropy to capture the probability of observed survival events. For patient $i$, $I_i$ is an indicator of event occurrence and $m_{i,k}$ encodes the event type $k$:

$$L_1 = \frac{1}{N} \sum_{i=1}^{N} \Big[ I_i \log_2 \Big( \sum_{k=1}^{K} m_{i,k} o_{i,k} \Big) + (1 - I_i) \log_2 \Big( \sum_{k=1}^{K} m_{i,k} o_{i,k} \Big) \Big]. \tag{25}$$

## B.2 Temporal Consistency Loss $L_2$

$L_2$ enforces temporal smoothness using a masked mean squared error, excluding missing variables (represented as $x_{i,t} = -\infty$) (Lee et al., 2019):

$$L_2 = \frac{1}{N} \sum_{i=1}^{N} \sum_{t=2}^{T} m_{i,t}(1 - m_{i,t})(y_{i,t} - x_{i,t})^2. \tag{26}$$

## B.3 CheXReport Regularization Loss $L_3$

To optimize multimodal feature extraction, we combine cross-entropy with double stochastic attention regularization (Xu et al., 2015). Given ground-truth sequence $y_{1:T}^*$ and predictions $y_t^*$ under parameters $\Theta$, we minimize:

$$L_3(\Theta) = - \sum_{t=1}^{T} \log_2 \big( p_\Theta(y_t^*|y_{t-1}^*) \big) + \sum_{l=1}^{L} \frac{1}{L} \sum_{d=1}^{D} \sum_{i=1}^{M^2} \Big( 1 - \sum_{c=1}^{T} \alpha_{ctdl} \Big), \tag{27}$$

where $\alpha_{ctdl}$ are attention weights, $D$ is the number of heads, and $L$ the number of layers. This regularization enforces more uniform attention distribution across spatial regions of the X-ray image.

## C COMPUTATIONAL COMPLEXITY

We now provide an analysis of the computational complexity of MultiTimeSurv. The total cost can be decomposed into three main parts: image encoding, temporal modeling, and multitask prediction.

**Image encoding.** The Swin Transformer encoder processes an $H \times W$ chest X-ray into $HW$ patches of dimension $d_v$. Since attention is restricted to local windows, the complexity is linear in the number of patches:

$$\Omega_{\text{image}} = O(HW \cdot d_v). \tag{28}$$

**Temporal modeling.** For a sequence of length $T$ with hidden size $d$, the GRU update cost is $O(Td^2)$, while the temporal attention mechanism adds $O(T^2 d)$. As $T$ is typically below 100 in clinical records, this term remains tractable:

$$\Omega_{\text{temporal}} = O(T^2 \cdot d). \tag{29}$$

**Multitask prediction.** Each of the $K$ risk-specific networks has $L$ dense layers of width $d$, resulting in:

$$\Omega_{\text{multitask}} = O(K \cdot L \cdot d^2). \tag{30}$$

**Overall.** Summing across components, the total time complexity of MultiTimeSurv is:

$$\Omega_{MultiTimeSurv} = O(HW \cdot d_v + T^2 \cdot d + K \cdot L \cdot d^2). \tag{31}$$

The memory cost is given by

$$O(HW + T \cdot |J| \cdot d + N_v \cdot d_v), \tag{32}$$

accounting for image patches, temporal embeddings for $|J|$ variables, and $N_v$ visual tokens in cross-attention.

In practice, these requirements remain tractable for typical clinical settings where $T < 100$ and $HW$ is moderate (e.g., downsampled X-rays). Thus, MultiTimeSurv balances expressivity with efficiency in real-world scenarios.

# D MYDIGITALHEALTH DATASET ANALYSIS

Between March 20, 2020, and June 02, 2022, we collected 1,891 hospitalization cases (1,815 unique patients) from Hospital de Clínicas de Porto Alegre (HCPA), Brazil. All patients were treated by the Brazilian public health system (SUS) and had a positive RT-qPCR test for SARS-CoV-2 at admission. Of these, 1,266 were admitted to the ICU. The dataset contains 36.57% deaths and 13.27% censored cases (patients transferred or who abandoned treatment).

Most cases originated from the state of Rio Grande do Sul (1,884 cases), with few from other states (SC, SP, RJ, RO, AM). The majority (1,193) were from Porto Alegre, reflecting the hospital's role as a regional reference center (Figure 3).

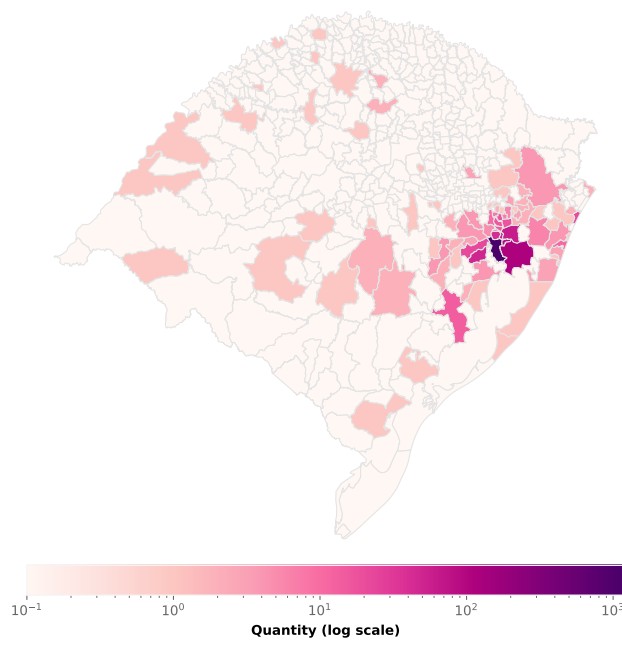

Figure 3: Patient stratification by city of origin in Rio Grande do Sul.

Table 3 summarizes sociodemographic information. The mean age was 59.15 years (IQR 48.0–71.0), with most cases in the 60–69 age group (25.54%). Male patients represented 51.51% of the cohort, with slightly higher mortality compared to females (28.34% vs. 27.58%). The majority self-identified as White (82.21%). Mortality was higher among Asian patients (45.46%) and decreased with higher educational attainment. Admission triage showed most patients were classified as very urgent (46.06%), with the highest mortality in emergency cases (45.00%).

Laboratory tests were available for 776 patients, yielding 15,289 collection records. Frequently collected tests included electrolytes (potassium, sodium), renal function markers (creatinine, urea, CKD-EPI, MDRD), and complete blood count parameters (Table 4).

In addition, 1,066 chest X-ray images were collected from 677 patients. Two detectors were used (DRX-1, 202 images; DRXPLUS3543C, 561 images), while 303 lacked metadata. Most exams were acquired in AP view (987 images), followed by PA (66) and lateral (13).

Finally, patients were temporally aligned by admission date. The time-to-event ranged from 0 to 209 days (IQR 5.0–23.0). Few samples had time-to-event greater than 50 days (Figure 4).

## D.1 SURVIVAL ANALYSIS METRICS

In survival analysis, risk prediction supports personalized treatment decisions (Pencina & D'Agostino, 2015). Model evaluation commonly relies on the Concordance index (C-index), which

Table 3: Patient characteristics stratified by outcome.

|  | General | Discharged | Death | Censored |
|---|---|---|---|---|
| **Age, in years** |  |  |  |  |
| Min | 18 | 18 | 21 | 20 |
| Max | 102 | 95 | 102 | 94 |
| Mean | 59.15 | 56.22 | 65.00 | 59.94 |
| Median | 61 | 57 | 67 | 61 |
| Standard deviation | 15.94 | 15.65 | 15.12 | 15.45 |
| Interquartile range | 48.0 - 71.0 | 44.0 - 67.0 | 57.0 - 75.0 | 50.0 - 70.0 |
| **Age groups, in years** |  |  |  |  |
| 18 to 29 | 74 (3.91%) | 50 (4.46%) | 15 (2.84%) | 9 (3.72%) |
| 30 to 39 | 181 (9.57%) | 138 (12.32%) | 27 (5.10%) | 16 (6.61%) |
| 40 to 49 | 261 (13.80%) | 194 (17.32%) | 35 (6.62%) | 32 (13.22%) |
| 50 to 59 | 373 (19.73%) | 232 (20.71%) | 85 (16.07%) | 56 (23.14%) |
| 60 to 69 | 483 (25.54%) | 275 (24.55%) | 141 (26.65%) | 67 (27.69%) |
| 70 to 79 | 330 (17.45%) | 154 (13.75%) | 137 (25.90%) | 39 (16.12%) |
| 80 ¡ | 189 (9.99%) | 77 (6.88%) | 89 (16.82%) | 23 (9.50%) |
| General | 1 891 | 1 120 (59.23%) | 529 (27.97%) | 242 (12.80%) |
| **Sex** |  |  |  |  |
| Female | 917 (48.49%) | 546 (48.75%) | 253 (47.83%) | 118 (48.76%) |
| Male | 974 (51.51%) | 574 (51.25%) | 276 (52.17%) | 124 (51.24%) |
| **Self-reported race** |  |  |  |  |
| Asian | 11 (0.58%) | 4 (0.36%) | 5 (0.95%) | 2 (0.83%) |
| White | 1 548 (82.21%) | 910 (81.61%) | 432 (81.82%) | 206 (85.83%) |
| Brown | 59 (3.13%) | 38 (3.41%) | 15 (2.84%) | 6 (2.50%) |
| Black | 265 (14.07%) | 163 (14.62%) | 76 (14.39%) | 26 (10.83%) |
| **Scholarity** |  |  |  |  |
| Illiterate | 61 (3.23%) | 24 (2.14%) | 24 (4.54%) | 13 (5.37%) |
| Elementary School | 580 (30.67%) | 331 (29.55%) | 186 (35.16%) | 63 (26.03%) |
| Middle School | 407 (21.52%) | 236 (21.07%) | 115 (21.74%) | 56 (23.14%) |
| High School | 497 (26.28%) | 341 (30.45%) | 101 (19.09%) | 55 (22.73%) |
| College / University | 117 (6.19%) | 75 (6.70%) | 29 (5.48%) | 13 (5.37%) |
| Without information | 229 (12.11%) | 113 (10.09%) | 74 (13.99%) | 42 (17.36%) |
| **Condition at admission** |  |  |  |  |
| Emergency | 20 (1.06%) | 10 (0.89%) | 9 (1.70%) | 1 (0.41%) |
| Very urgent | 871 (46.06%) | 589 (52.59%) | 182 (34.40%) | 100 (41.32%) |
| Urgent | 124 (6.56%) | 96 (8.57%) | 20 (3.78%) | 8 (3.31%) |
| Less urgent | 7 (0.37%) | 6 (0.54%) | 0 (0.00%) | 1 (0.41%) |
| Not classified | 869 (45.95%) | 419 (37.41%) | 318 (60.11%) | 132 (54.55%) |

Table 4: Percentage of patients exams results available in the HCPA sample.

| Exam | % | Exam | % | Exam | % |
|---|---|---|---|---|---|
| Potassium | 25.51 | Sodium | 25.84 | Serum creatinine | 30.37 |
| Urea | 30.46 | CKD-EPI | 30.52 | MDRD | 30.54 |
| Magnesium | 57.96 | Leukocytes | 68.98 | Monocytes % | 68.98 |
| Absolute monocytes | 68.98 | Absolute segmented neutrophils | 68.98 | Segmented neutrophils % | 68.98 |
| Hematocrit | 68.98 | Hemoglobin | 68.98 | Absolute lymphocytes | 68.98 |
| Lymphocytes % | 68.98 | Absolute eosinophils | 68.99 | Absolute basophils | 68.99 |
| Basophils % | 68.99 | Eosinophils % | 68.99 | MCV Failace | 68.99 |
| MCH | 68.99 | MCHC | 68.99 | Erythrocytes | 68.99 |
| RDW | 69.00 | Erythroblasts | 69.02 | C-reactive protein result | 72.08 |
| H obs1 blood count | 80.70 | APTT control | 81.39 | APTT seconds | 81.39 |
| Blood count observation neutrophil bands | 83.12 | Calcium VR | 88.42 | Corrected calcium | 88.44 |
| Absolute band neutrophils | 88.59 | Band neutrophils % | 88.59 | Absolute myelocytes | 89.52 |
| Myelocytes % | 89.52 | H obs2 blood count | 89.68 | PT control | 91.75 |
| PT INR | 91.75 | PT seconds | 91.75 | PT activity | 91.75 |
| H D-dimers | 93.03 | Serum chloride | 93.09 | Direct bilirubin | 93.37 |
| Indirect bili result | 93.37 | Total bilirubin | 93.39 | CK | 93.43 |
| Metamyelocytes % | 93.81 | Absolute metamyelocytes | 93.81 | GPT | 94.38 |
| GOT/AST result | 94.42 | Plasma lactate | 94.72 | H obs3 blood count | 95.08 |
| Troponin-T | 95.70 | LDH VR | 96.08 | STA compact fibrinogen | 97.32 |
| H obs4 blood count | 97.90 | Albumin | 98.16 | Observation | 98.26 |
| Estimated average glucose | 98.79 | A1C | 98.79 | Plasma cells % | 98.86 |
| Absolute plasma cells | 98.86 | Ferri result | 99.18 | H obs5 blood count | 99.29 |
| Absolute promyelocytes | 99.30 | Promyelocytes % | 99.30 | Minor/major indicator | 99.37 |
| Triglycerides | 99.44 | Sample creatinine | 99.48 | Absolute reticulocytes | 99.57 |
| Reticulocytes | 99.57 | Urine sample sodium | 99.62 | H obs6 blood count | 99.67 |
| Biochemical observations | 99.67 | E170 signal | 99.69 | Indicator seconds | 99.74 |
| Obs | 99.76 | Sample urea | 99.84 | CKD-EPI alpha | 99.86 |
| Urine sample potassium | 99.88 | C3 result | 99.89 | C4 result | 99.89 |
| Rheumatoid factor result | 99.91 | Result | 99.93 | CD3 value | 99.95 |
| H IF CD45/UL | 99.95 | H CD8 % | 99.95 | H CD4 % | 99.95 |
| CD4/CD8 ratio | 99.95 | CD4 value | 99.95 | CD8 value | 99.95 |
| H CD3 % | 99.95 | CSF lactate | 99.95 | Urine sample chloride | 99.95 |
| IgG result | 99.95 | Mean fluorescence index (MFI) | 99.97 | Thrombin time numeric | 99.97 |
| H obs7 blood count | 99.97 | IgM result | 99.97 | CSF ADA support | 99.97 |
| Urine volume | 99.98 | CSF LDH | 99.98 | Iron | 99.99 |
| 24h creatinine | 99.99 | Urine alpha calcium | 99.99 | Magnesium result | 99.99 |
| 24h calcium | 99.99 | ADA support | 99.99 | Urine alpha creatinine | 99.99 |
| CEA result | 99.99 | Ascites albumin | 99.99 | Absolute blasts | 99.99 |
| Blasts % | 99.99 | Urine sample calcium | 99.99 | Urinary urea | 99.99 |
| APTT obs | 99.99 | Activity indicator | 99.99 | INR indicator | 99.99 |
| Total ascites bili 1 | 99.99 | CD4 observation | 99.99 | Total potassium volume | 99.99 |
| Urine alpha sodium | 99.99 | Serous fluid ADA | 99.99 | Thrombin time signal | 99.99 |
| Biochemical results outside measurement range | 99.99 | Urine alpha urea | 99.99 | Urine alpha potassium | 99.99 |
| 24h urine sodium | 99.99 | Calcitonin result | 100.00 | Antithrombin result | 100.00 |
| FO urea | 100.00 | | | | |

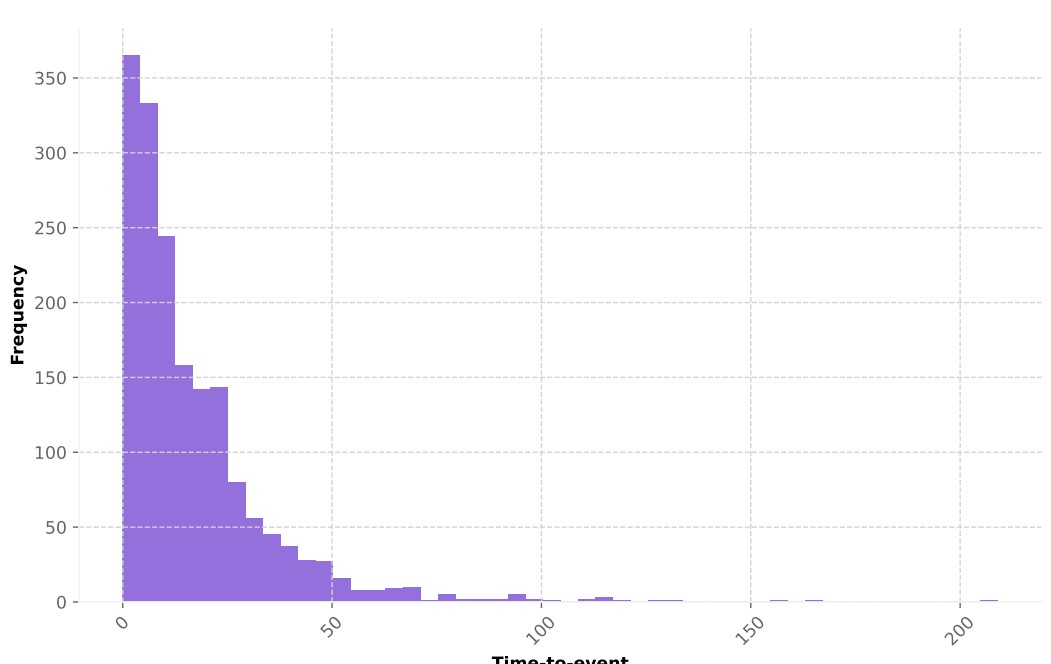

Figure 4: Time-to-event distribution in the dataset.

measures the proportion of correctly ordered patient pairs with respect to observed event times (Park et al., 2021). Values range from 0.5 (random ordering) to 1.0 (perfect discrimination). Formally, the C-index is defined as

$$\text{C-index} = \frac{\sum_{i,j} I(h_i > h_j)\, I(t_i < t_j)\, \varphi_i}{\sum_{i,j} I(t_i < t_j)}, \tag{33}$$

where $h_i$ and $h_j$ denote predicted risk scores, $t_i$ and $t_j$ the observed survival times, $I(\cdot)$ an indicator function, and $\varphi_i$ indicates whether the event was observed $(1)$ or censored $(0)$.

Beyond discrimination, overall predictive accuracy can be assessed using the Brier score (Park et al., 2021), which measures the mean squared error between predicted survival probabilities and observed outcomes:

$$\text{Brier score} = \frac{1}{N} \sum_{i=1}^{N} (p_i - o_i)^2, \tag{34}$$

where $N$ is the number of observations, $p_i$ the predicted event probability, and $o_i \in \{0, 1\}$ the observed outcome. Lower values indicate better calibration, with $0$ representing perfect prediction.

### D.2 HYPERPARAMETER SEARCH SPACE

Hyperparameters for the MultiTimeSurv training were optimized using Random Search. The ranges explored are summarized in Table 5.

Table 5: Search space for MultiTimeSurv network hyperparameters.

| Hyperparameter | Value Space |
|---|---|
| Batch size | 32 |
| Dense layers activation | {ReLU, ELU, tanh} |
| RNN activation | {ReLU, ELU, tanh} |
| Dense output activation | {ReLU, ELU, tanh} |
| Dense dropout | {0.2, 0.4, 0.6} |
| RNN dropout | {0.2, 0.4, 0.6} |
| RNN cell | {GRU, LSTM} |
| RNN hidden size | {25, 50, 100, 150, 200} |
| Dense hidden size | {25, 50, 100, 150, 200} |
| Dense number of layers | {1, 2, 3} |
| RNN dense number of layers | {1, 2, 3} |
| Attention number of layers | {2, 4, 6, 8, 10} |
| Learning rate | {1e-3, 1e-4, 1e-5} |
| Embeddings dropout | {0.2, 0.4, 0.6} |
| Embeddings size | {8, 16, 32, 64} |

For the CheXReport network, we performed an architecture evaluation using the MIMIC-CXR dataset. We compared Swin-T, Swin-S, and Swin-B encoders pre-trained on ImageNet. The number of encoder layers was fixed to $N_d = 2$, and the visual embedding dimension was set to 256. Multilingual BERT embeddings were frozen for the first 10 epochs to allow the image encoder to adapt to the medical domain. During inference, report generation used a beam size of 5, stopping at the <eos> token or when reaching the maximum token length.

We also evaluated traditional feature extraction pipelines for image captioning. Specifically, we compared ResNet50-v2 and ResNet101-v2 encoders combined with two decoders: (i) fusion of visual features with BERT embeddings, and (ii) the CheXReport decoder. These models were pre-trained for 100 epochs with Adam (learning rate $1e^{-4}$, reduced by 25% after 10 epochs without BLEU-4 improvement). Batch size was fixed to 64. All experiments were implemented in Python using PyTorch and OpenCV, and executed on a system with a 24GB Quadro RTX 6000 GPU, Intel Xeon Silver 4216 CPU @ 2.10GHz, and 64GB RAM.

### D.3 BASELINES

We benchmarked MultiTimeSurv against established survival analysis architectures: CoxTime, CoxCC, DeepSurv, PCHazard, DeepHit, and N-MTLR.

- **CoxTime** extends Cox proportional hazards by incorporating neural networks to capture non-linear effects over time (Kvamme et al., 2019).
- **CoxCC** combines case-control sampling with neural networks for improved efficiency and accuracy (Kvamme et al., 2019).
- **DeepSurv** is a deep learning adaptation of the Cox model designed for time-to-event prediction (Katzman et al., 2018).
- **PCHazard** partitions time into intervals and estimates hazards piecewise via neural networks, allowing temporal variation in risk (Kvamme et al., 2019).
- **DeepHit** directly models the time-to-event distribution as a multi-class classification problem (Lee et al., 2018).
- **N-MTLR** employs neural multi-task logistic regression to estimate the survival function, capturing temporal dependence and covariate interactions (Fotso, 2018).

### D.4 PRE-PROCESSING

Data used in survival analysis is heterogeneous across hospital systems. Clinical variables are often stored in Clinical Information Systems (CIS), images in PACS, and biomarkers in Laboratory Information Systems (LIS). Differences in data formats, collection protocols, and measurement units necessitate a dedicated pre-processing pipeline.

For each patient $i$, we define a dataset $D_i = \{(X_i, \rho_i, \delta_i)\}_{i=1}^N$, where $N$ is the number of patients, $X_i = (C_i, M_i, I_i)$ is the feature tuple at time $t \in \tau = \{1, \ldots, T\}$, $\rho_i$ is the survival time, and $\delta_i$ is the event indicator. The covariate set $C_i = \{c_{i,1}, \ldots, c_{i,j}\}$ includes categorical and continuous variables. To represent missing values, we assign $x_{i,j} = -\infty$ and provide a mask $M = \{m_{i,1}, \ldots, m_{i,j}\}$ defined as

$$m_{i,j} = \begin{cases} 1, & \text{if } x_{i,j} = -\infty, \\ 0, & \text{otherwise.} \end{cases} \tag{35}$$

The event indicator is defined as

$$\delta = \begin{cases} k, & \text{if patient } i \text{ experienced cause } k \in K = \{1, \ldots, K\}, \\ 0, & \text{if censored.} \end{cases} \tag{36}$$

Continuous covariates are standardized to zero mean and unit variance:

$$J_{(cont),z} = \frac{c_z - u_z}{\sigma_z}, \tag{37}$$

where $u_z$ and $\sigma_z$ are the mean and standard deviation of variable $z$.

**Image Pre-processing.** Each chest X-ray image $\iota$ with height $h$ and width $w$ is resized to $\omega \times \omega$. Padding of size $\gamma = \max(h, w) - \min(h, w)$ is applied along the smaller axis:

$$(h', w') = \begin{cases} (h + \gamma, w), & \text{if } h < w, \\ (h, w + \gamma), & \text{if } h > w, \\ (h, w), & \text{otherwise.} \end{cases} \tag{38}$$

To mitigate contrast variability, we apply Contrast Limited Adaptive Histogram Equalization (CLAHE). An image is partitioned into blocks $B$ of size $u \times v$. For each block $B_{a,b}$, the histogram $H_{a,b}(z)$ of pixel intensities $z$ is clipped with threshold

$$\psi = \alpha \frac{u \times v}{L}, \tag{39}$$

where $\alpha$ is the clipping factor and $L$ is the number of intensity levels. The clipped histogram is

$$H'_{a,b}(z) = \min(H_{a,b}(z), \psi), \qquad (40)$$

and redistributed as

$$H''_{a,b}(z) = \frac{H'_{a,b}(z) + \sum_z (H_{a,b}(z) - \psi)}{L}. \qquad (41)$$

The transformation function is

$$F_{a,b}(z) = \frac{L-1}{uv} \sum_{l=0}^{z} H''_{a,b}(l), \qquad (42)$$

applied to each pixel $p$ as $p' = F_{a,b}(p)$. For pixels at block edges, bilinear interpolation ensures smooth transitions.

## E  RESULTS ON PBC2

PBC2 provides a unimodal benchmark with longitudinal tabular covariates. As shown in Table 6, MultiTimeSurv achieves the best performance despite not leveraging imaging, with C-index improvements of $+0.100$ over CoxPH and $+0.03$ over Dynamic-DeepHit. This demonstrates that the inductive biases for irregular sampling and temporal embeddings generalize beyond multimodal contexts.

Table 6: Comparison of algorithms at different prediction times $t$ and horizons $\Delta t$ for respiratory failure. Values are mean $\pm$ std.

| $t$ | Algorithms | $\Delta t = 1$ | $\Delta t = 3$ | $\Delta t = 5$ | $\Delta t = 10$ |
|---|---|---|---|---|---|
| | CoxPH | 0.840±0.09† | 0.837±0.08† | 0.837±0.08† | 0.837±0.08† |
| | RSF | 0.931±0.02* | 0.931±0.02* | 0.929±0.01* | 0.927±0.01* |
| $t = 30$ | JM | 0.896±0.04* | 0.897±0.04* | 0.897±0.04* | 0.897±0.04* |
| | JM-LC | 0.897±0.04* | 0.897±0.04* | 0.897±0.04* | 0.897±0.04* |
| | DynamicDeepHit | 0.948±0.01 | 0.939±0.01 | 0.938±0.01 | 0.937±0.01 |
| | **MultiTimeSurv** | **0.951 ± 0.01** | **0.943 ± 0.00** | **0.944 ± 0.01** | **0.941 ± 0.02** |
| | CoxPH | 0.887±0.09* | 0.887±0.09* | 0.887±0.09* | 0.887±0.09* |
| | RSF | 0.888±0.10* | 0.887±0.10* | 0.887±0.10* | 0.887±0.10* |
| $t = 40$ | JM | 0.911±0.04† | 0.913±0.04† | 0.913±0.04† | 0.913±0.04† |
| | JM-LC | 0.911±0.04† | 0.913±0.04† | 0.913±0.04† | 0.913±0.04† |
| | DynamicDeepHit | 0.956±0.01 | 0.956±0.01 | 0.956±0.01 | 0.958±0.01 |
| | **MultiTimeSurv** | **0.969 ± 0.01** | **0.967 ± 0.00** | **0.966 ± 0.01** | **0.961 ± 0.02** |
| | CoxPH | 0.851±0.11* | 0.851±0.11* | 0.851±0.11* | 0.851±0.11* |
| | RSF | 0.896±0.10* | 0.896±0.10* | 0.896±0.10* | 0.896±0.10* |
| $t = 50$ | JM | 0.919±0.04* | 0.919±0.04* | 0.919±0.04* | 0.919±0.04* |
| | JM-LC | 0.919±0.04* | 0.919±0.04* | 0.919±0.04* | 0.919±0.04* |
| | DynamicDeepHit | 0.962±0.01 | 0.962±0.01 | 0.962±0.01 | 0.961±0.01 |
| | **MultiTimeSurv** | **0.978 ± 0.01** | **0.975 ± 0.00** | **0.974 ± 0.01** | **0.973 ± 0.02** |

## F  LIMITATIONS AND FUTURE DIRECTIONS

The MultiTimeSurv model presents several limitations. First, the dataset originates from a single institution, which constrains generalizability. Its demographic and regional homogeneity may restrict applicability in settings with different populations or healthcare practices. Second, model performance is dependent on the quality and completeness of EHRs and imaging data. Despite employing mechanisms for handling missing values, inconsistencies, and sparsity, uncertainty and potential bias are introduced. Third, the reliance on chest X-ray images and structured clinical data only partially reflects the multifactorial nature of COVID-19. The absence of richer modalities (e.g., CT

scans, longitudinal patient histories, genomic data) limits predictive scope. Moreover, interpretability remains challenging: although explainability components were incorporated, the underlying deep models retain black-box characteristics, which may hinder clinician trust and adoption.

The evaluation was retrospective, highlighting the need for prospective validation in diverse settings to confirm its utility in real-time decision-making. The clinical impact of MultiTimeSurv on outcomes, resource allocation, and patient management also remains unassessed. In addition, while robust for short- and medium-term horizons, long-term prediction capabilities were not systematically evaluated. Given the dynamic evolution of patient health trajectories and treatment protocols, continual model updates will be required to sustain accuracy.

Future work should address these issues by incorporating multi-institutional and multi-regional datasets, integrating additional modalities (e.g., CT, genomics), and conducting prospective studies to assess real-world effectiveness. Enhancing transparency and interpretability will be essential for clinical integration. Furthermore, adaptive and continual learning strategies (e.g., transfer learning, online learning) should be explored to ensure resilience to evolving data distributions. Finally, ethical and privacy considerations remain critical: mitigating biases, protecting patient data, and establishing governance frameworks for responsible AI deployment will be necessary for broader acceptance and trust.

