# OpenReview forum: "MultiTimeSurv: Temporal Multimodal Networks for Dynamic Survival Analysis with Longitudinal Data"
_ICLR.cc/2026/Conference — Submitted to ICLR 2026_

### Official Review · Reviewer_v42K · 2025-10-15

**Soundness:** 3
**Presentation:** 2
**Contribution:** 2
**Rating:** 4
**Confidence:** 2

**Summary:**

This paper introduces MultiTimeSurv, a novel deep learning framework designed for dynamic survival analysis using irregularly sampled, multimodal longitudinal data. The model specifically tackles the integration of time-series tabular data (clinical and lab values) with sparsely available medical images and their associated text reports. Its architecture has three core components: (1) sophisticated feature encoders for heterogeneous tabular data that capture cyclical and distributional patterns; (2) an attention-based recurrent network (GRU) to model temporal evolution from irregularly sampled data points; and (3) a fully transformer-based module (CheXReport) to jointly process and align images with textual reports. By fusing these rich representations, MultiTimeSurv performs multitask learning to predict the risk of competing events over time.

**Strengths:**

1. Unified and Comprehensive Framework: The primary strength of MultiTimeSurv is its ability to holistically address the key challenges of real-world longitudinal survival analysis in a single, end-to-end framework. It simultaneously handles heterogeneous data types (categorical/continuous), irregular temporal sampling, competing risks, and multimodal fusion (tabular, image, and text), which is a significant step beyond prior models that typically focus on only a subset of these problems.

2. Sophisticated Feature Engineering for Tabular Data: The paper proposes an advanced method for embedding continuous tabular variables, combining periodic, piecewise, and distributional strategies.

3. Advanced Multimodal Fusion: The integration of the CheXReport architecture for jointly encoding and aligning medical images with their corresponding text reports is a major strength.

**Weaknesses:**

1. Fairness of comparison: This model utilizes more multimodal data and achieves better results. The ablation experiment can provide the results of removing one modal branch.

2. Lack of corresponding ablation experiments.

3. Dependence on High-Quality, Paired Data: The effectiveness of the CheXReport module relies on the availability of image-text pairs. In many clinical settings, imaging data may exist without detailed, structured reports, or vice-versa.

4. Scalability of the Multitask Framework: The model uses a separate neural network for each of the K competing risks. How does this approach scale computationally and in terms of performance as the number of competing events grows?

**Questions:**

1. Fairness of comparison: This model utilizes more multimodal data and achieves better results. The ablation experiment can provide the results of removing one modal branch.

2. Lack of corresponding ablation experiments.

3. Dependence on High-Quality, Paired Data: The effectiveness of the CheXReport module relies on the availability of image-text pairs. In many clinical settings, imaging data may exist without detailed, structured reports, or vice-versa.

4. Scalability of the Multitask Framework: The model uses a separate neural network for each of the K competing risks. How does this approach scale computationally and in terms of performance as the number of competing events grows?

---

### Official Review · Reviewer_FW2u · 2025-10-25

**Soundness:** 2
**Presentation:** 2
**Contribution:** 2
**Rating:** 4
**Confidence:** 2

**Summary:**

The paper introduces MultiTimeSurv, a unified deep learning framework for temporal multimodal survival analysis.
It integrates longitudinal tabular data (categorical + continuous features with missing values) and imaging data (chest X-rays + textual reports) for dynamic time-to-event prediction with censoring and competing risks. Key contributions include (1) Periodic, piecewise, and distributional embeddings for continuous and categorical variables. (2) Handles irregular sampling and missingness in longitudinal data. (3) Uses a transformer-based CheXReport encoder–decoder to align visual and textual information. (4) Predicts multiple risk types simultaneously. Experiments on MDH and SYMILE-MIMIC datasets show significant gains in C-index and Brier score over baselines like DeepHit and Dynamic-DeepHit.

**Strengths:**

1. Experiments on two multimodal clinical datasets with consistent superiority in both discrimination (C-index > 0.70) and calibration. Ablation studies support each design choice (embeddings, attention, multimodal fusion).
2. Implementation details, loss functions, and complexity analysis are thoroughly described. A GitHub repository is promised, which is positive for reproducibility.
3. The combination of periodic/piecewise embeddings, temporal attention, and multimodal fusion is well-motivated.

**Weaknesses:**

1. Limited novelty in architecture combination. The model integrates several known components (GRU + attention + Swin Transformer + multitask head), mainly combining existing ideas rather than introducing fundamentally new mechanisms.
2. While ablations confirm each component’s contribution, the paper does not isolate cross-modal attention mechanisms or analyze how multimodal fusion improves interpretability beyond quantitative metrics.

**Questions:**

You handle missing covariates by masking and assigning a value of −∞. How robust is this approach when missing rates become higher?

---

### Official Review · Reviewer_8Tur · 2025-10-30

**Soundness:** 2
**Presentation:** 1
**Contribution:** 2
**Rating:** 2
**Confidence:** 3

**Summary:**

MultiTimeSurv is a survival model which incorporates multiple data modalities of time series, namely continuous and categorical tabular values and chest X-rays. The authors use high-dimensional learned latent embeddings of these data modalities as input to a recurrent network with an attention mechanism to learn temporal dependencies. The authors explicitly mask missing values to circumvent the need for data imputation. The hidden states of the recurrent network and the embedded data are then used to predict the probability of survival events. The authors apply their model to two clinical datasets and achieve C-index ~0.7 and Brier-score ~0.1-0.2.

**Strengths:**

- Well motivated and well founded problem addressing the realities and complexities of survival modeling with messy clinical data that has variable sampling and missing data..

- Laudable effort to do evaluation of their method over multiple datasets: SYMILE-MIMIC and MDH, the latter which seems to be a multi-institutional private dataset from Brazil. This is further commendable to the authors for investing the time and effort to collect this data.

- Nice set of comparisons to other methods from the literature in the main results. Helps contextualize the potential benefit of MultiTimeSurv relative to these other methods.

**Weaknesses:**

- Poor presentation of equations and symbols. In Section 3.1, lots of implied or otherwise undefined variables (e.g. Y_i, T, K, \iota, \epsilon. Also some redefinition of symbols in Figure 1 i.e. c_{i,t}. Section 3.2 line 239-243, authors discuss time-varying missingness but drop time notation in explanation of the mask, leading to ambiguity.

- It is extremely unclear why Section 3.4 details the use of CheXReport for radiology report generation. Reading Appendix Section B.3 seems to imply that the radiology report generation is a guiding task in the composite loss function that helps derive meaningful features, but nowhere is this stated in Section 3.4. Furthermore, it is implied earlier in the introduction and preliminaries that paired reports may serve as some other input to the model, yet that does not seem to be the case.

- The authors describe how they aim to model irregularly measured data. There is nuance that is lost in whether they mean measurements at imprecise intervals or sparsely sampled measurements. It is unclear how relative temporal information is encoded in the former case, unless the authors are aligning measurements to discrete time intervals (e.g. hours). It seems perhaps the authors are doing time alignment, as on line 330 they discuss the experimental setup with seemingly discrete evaluation times. It is also unclear how chest x-rays prior to the prediction time are incorporated, as the per-class classifiers are used at every time step. This lack of clarity is a major weakness of the work.

- The authors only use a subset of data modalities from the SYMILE-MIMIC dataset. It is unclear why the method was not generalized to include the ECGs included in SYMILE as there are many foundational ECG models for extracting rich embeddings.

- The discussion presented in section 4.3 is severely lacking. If anything, Patient C demonstrates a very serious false negative for patient mortality. The author’s point about the intersection of risk would make sense if the patient has passed away during the ambiguous period (looks like maybe days 11-16), but the model predicted probabilities clearly diverged into a false negative. Also it is unclear why the probabilities do not sum to 1.

- No mention of what the baseline models A or B are. Needs further explanation beyond the provided footnote on lines 368/369.

**Questions:**

- GitHub repository linked in reproducibility statement is empty

- Line 125/6, link to overview figure is misnumbered

- Notation on lines 143-147 seems poorly formatted, was this supposed to be a bulleted list?

- It is unclear if the K events are mutually exclusive or if this is a multilabel problem. The joint softmax at each time step (eq 19) seems to imply so. Should be explicitly stated.

- The loss components introduced in Equation 20 should not be relegated to the appendix.

- Should clarify that MDH is your data and reference Appendix D

- SYMILE-MIMIC should be cited

---

### Official Review · Reviewer_Dgid · 2025-10-31

**Soundness:** 2
**Presentation:** 2
**Contribution:** 2
**Rating:** 2
**Confidence:** 2

**Summary:**

MultiTimeSurv presents a deep learning framework for multimodal survival analysis, attempting to solve challenges in processing heterogeneous, longitudinal medical data. The model tries to address computational challenges of irregular sampling, missing data, and cross-modal interactions through an integrated approach using temporal attention mechanisms, specialized feature encoders, and transformer-based architectures. The research explores survival prediction across multiple datasets, with an emphasis on medical prognosis scenarios, and provides a methodology for analyzing time-to-event outcomes through multimodal data integration.

**Strengths:**

The paper shows no apparent grammatical errors or factual inaccuracies.

**Weaknesses:**

1.	The proposed methodology lacks innovation, with attention mechanisms and RNN techniques being classical approaches for addressing general problems and understanding medical data. The method offers no substantive inspiration for the field.
2.	The method section is presented in an overly naive manner, failing to meet the standards of a serious academic manuscript. The exposition lacks the depth and scholarly rigor expected in a high-caliber research paper.
3.	The figures, tables, and overall typesetting appear unprofessional and do not conform to ICLR's publication standards.

**Questions:**

Please refer to weakness.

---

### Meta-Review · Area_Chair_DDCB · 2026-01-01

**Summary:**

The reviewers acknowledged the importance of the problem and the empirical performance on multiple clinical datasets, but raised consistent concerns about limited methodological novelty, as the approach mainly combines existing components. More critically, they identified substantial issues with clarity and presentation, including poorly defined notation, ambiguous treatment of irregular sampling and multimodal fusion, and unclear experimental setup and baselines. Additional concerns about reproducibility and incomplete code release further weakened confidence, leading to the recommendation to reject.

**Reviewer Concerns:**

There is no rebuttal.

**Reviewer Scores:**

N/A

---

### Decision · Program_Chairs · 2026-01-26

Reject